# Forwarder Productivity in Salvage Logging Operations in Difficult Terrain

**Alberto Cadei [1], Omar Mologni [2] , Dominik Röser [2], Raffaele Cavalli [1] and Stefano Grigolato [1,\***

[1] Department of Land, Environment, Agriculture and Forestry, Università degli Studi di Padova, viale dell'Università 16, 35020 Legnaro, Padova, Italy; alberto.cadei@phd.unipd.it (A.C.); raffaele.cavalli@unipd.it (R.C.)

[2] Department of Forest Resources Management, Faculty of Forestry, The University of British Columbia, MainMall 2424, Vancouver, BC V6T 1Z4, Canada; omar.mologni@ubc.ca (O.M.); dominik.roeser@ubc.ca (D.R.)

\* Correspondence: stefano.grigolato@unipd.it

**Abstract:** Large scale windthrow salvage logging is increasing in Central Europe because of the growth of severe atmospheric events due to global heating. Sustainable forest operations in these conditions are challenging in terms of both productivity performances and safety of the operations. Fully mechanized harvesting systems are the preferred solution on trafficable terrains and proper slopes. However, different work methods and logistic organization of the operations could largely change the overall performances. The study observed three harvesting sites based on fully mechanized cut-to-length systems and located in areas affected by the Vaia storm, which hit north-eastern Italy in October 2018. The objectives were to estimate forwarder productivity in salvage logging in difficult terrain and to identify significant variables affecting this productivity under real working conditions. Time and motion studies were carried out and covered 59.9 $PMH_{15}$, for a total of 101 working cycles, extracting a total volume of 1277 $m^3$ of timber. Average time consumption for each site was 38.7, 42.2, and 25.1 $PMH_{15}$ with average productivity of 22.5, 18.5, and 29.4 $m^3/PMH_{15}$, respectively, for Sites A, B, and C. A total of seven explanatory variables significantly affected forwarder productivity. Average load volume, maximum machine inclination during loading, and number of logs positively affected the productivity. On the contrary, travel distance, load volume, maximum ground slope during moving and loading have a negative influence. With an average travel distance of 500 m, the productivity resulted 20.52, 16.31, and 23.03 $m^3/PMH_{15}$, respectively, for Sites A, B, and C. An increase of 200 m of travel distance causes a decrease in productivity of 6%.

**Keywords:** efficiency; steep terrain; cut-to-length; windthrow; Vaia storm

## 1. Introduction

Forest stands cover 43% of the European land and provide significant socioeconomic values through their ecosystem services. The European Commission confirms once again the fundamental role of forests in its strategic long-term vision for a prosperous, competitive, and climate neutral economy [1]. This strategy can be seriously compromised due to the increment of forest disturbance in terms of frequency, severity, and extent mainly due to global heating [2]. Drought related losses, the increase in intensity and frequency of threats (e.g., wildfires and wind storms), and the subsequent exponential increase in biological risks (e.g., bark beetle outbreaks) are the most significant natural disturbances currently affecting the dynamics of forest ecosystems in Europe [3,4] and the world in general [5]. As a result, large post-disturbance management strategies focus on (i) active interventions

or on (ii) passive management [6]. Active intervention strategies focus on rapid post-disturbance harvest and recovery of the economic value of the forest [7] in order to decrease the risk of a rapid reduction of the timber value due to reductions in wood quality [8], the risk of wildfires [9], and insect outbreaks [10].

A common post-disturbance management approach is salvage logging [11,12] which consists of the widespread removal of damaged trees. Salvage logging benefits and drawbacks are widely discussed as it can have a negative effect on forests in terms of reducing biodiversity, increasing erosion, and reducing soil fertility [11]. Some authors report that salvage logging interventions must be planned considering the site-specific characteristics [13,14]. When salvage logging is appropriate to be applied, the most suitable technological solutions are those based on fully mechanized systems as these guarantee high productivity and above all a lower risk for operators as they work exclusively on the machines [15,16].

In complex terrain and in mountain areas, with a low density of forest roads, the use of fully mechanized systems in salvage logging operations is difficult. In these conditions, the main system remains the semi-mechanized system based on motor-manual felling with a chainsaw and timber extraction by cable yarders or skidders [17]. In recent years, even the fully mechanized system based on the combined use of harvesters and forwarders has been spreading with the use of heavier and more suitable machines for working on steep and trafficable terrain [18,19]. Currently, harvesters with independently suspended tracks or wheels mounted on hydraulically driven arms or forwarders with heavy-duty portal bogie axles with balancing system and self-levelling cabins, and fitted with a synchronized winch to improve traction are the main developments introduced in ground-based harvesting system in the last years to operate in complex and steep terrain up to 75–85% slope [18]. Forwarders are machines typically used in fully mechanized cut-to-length (CTL) timber harvesting with the aim to extract timber from the forest to the landing sites [20,21]. Due to the lower risk for forest operators, lower harvesting costs, and higher productivity [22], fully mechanized CTL harvesting systems are commonly applied in mountain regions in order to harvest damaged trees from the forests.

Forwarder productivity has been widely studied in Europe. The main factors that affect forwarder productivity are related to forest stand and ground conditions such as density of the trees, volume of payload, number of logs extracted, number of log assortments, average log volume, extraction distances and direction of wood extraction, terrain slope, operator's experience [20,23–25]. Typically, slope and extraction distance have a negative influence on productivity while volume of payload has a positive influence [26]. According to Ghaffarian et al. [27] the extraction distance has a negative influence on productivity, while the increase of load volume and downhill slope have a positive influence on productivity. Payload should be positively related to productivity, Eriksson and Lindroos [28] find that the productivity between forwarders with different payloads decreased with extraction distance, both in final felling and thinning. Obviously, larger machines require a higher investment but, on the other hand, are expected to have higher productivity and a lower cost per cubic meter [29]. Additionally, type of operation and stand affected forwarder productivity, for example In Ireland, in clearcut and thinned working areas forwarder productivity could range from 13.57 to 27.25 m$^3$/PMH and from 7.28 to 13.92 m$^3$/PMH, respectively [20]. Previous studies mentioned productivity related to ordinary harvesting conditions or salvage logging operation in gentle terrain, while there is a lack of knowledge about the productivity of forwarders in non-ordinary conditions, such as salvage logging, in complex and steep terrain. For example, in Finland cutting costs in salvage logging operations were estimated to be about 35–64% higher than ordinary clearcutting and the logging costs were 10–30% higher than normal standing stems [15,17]. Although harvester productivity in salvage logging operations are lower, compared with normal clearcutting, Bergkvist [30] does not detect any significant difference in forwarder productivity in salvage logging operations compared to normal clearcutting. This paper aims to partially fill the lack of knowledge of forwarder productivity in salvage logging operation in difficult terrain. The specific objectives of the paper are (i) to estimate time consumption and evaluate forwarder productivity in salvage logging in difficult terrain and (ii) to identify the most significant

variables affecting productivity of forwarding operation in salvage logging operations in difficult terrain and under real working conditions.

## 2. Materials and Methods

### 2.1. Case Studies

The study was conducted in 2019 in the Italian Alps in the forest area affected by windthrow caused by Vaia storm that occurred at the end of October 2018 [31,32]. Data were collected in three different sites, respectively, located in Levico Terme (TN), Aldino (BZ), and Belluno (BL) (Site A, Site B, and Site C) (Figure 1) (Table 1).

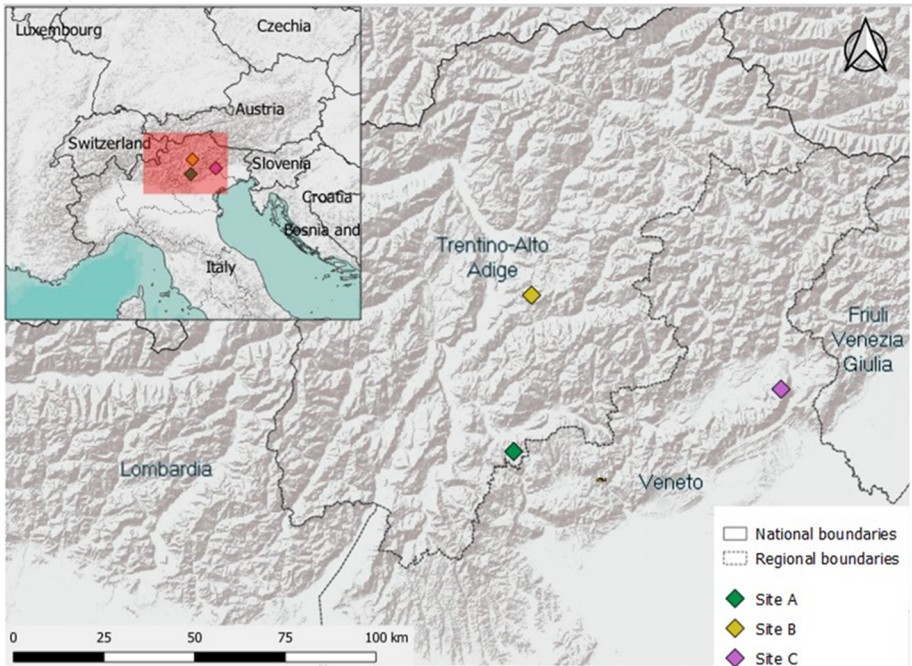

**Figure 1.** Location of the different working areas.

**Table 1.** Stand description and characteristics of damage for the three different working areas.

|  | Site A | Site B | Site C |
|---|---|---|---|
| Location | Passo Vezzena | Redagno di sopra | Nevegal |
| Province | Trento | Bolzano | Belluno |
| Elevation (m a.s.l.) | 1460 | 1600 | 1300 |
| Total area (ha) | 20 | 30.5 | 60.48 |
| Damaged area (%) | 65 | 100 | 40 |
| Estimated damaged wood (m$^3$) | 3500 | 10,000 | 25,000 |
| Average slope (%) | 17 | 24 | 33 |
| **Species (% of volume)** |  |  |  |
| *Abies alba* | 35 | 30 | - |
| *Picea abies* | 65 | 60 | 73 |
| *Larix decidua* | - | 10 | 21 |
| *Broad leaved* | - | - | 6 |

Site A was composed of a mixed even-aged stand with spruce (*Picea abies*) and silver fir (*Abies alba*) located at an average altitude of 1460 m a.s.l. The damaged area was estimated to be about 65% of the total stand area. Site B was composed by a mixed even-aged stand with spruce, silver fir, and European larch (*Larix decidua*) located at an average altitude of 1600 m a.s.l. In this site, the damaged area was

estimated to be about 100% of the stand. Site C was composed of a mixed uneven-aged stand with spruce and European larch, with different broad-leaved species such as beech (*Fagus sylvatica*), maple (*Acer pseudoplatanus*), and birch (*Betula pendula*) mixed in. Overall, the broad-leaf volume did not exceed 6% of the total estimated volume. Site C was located at an average altitude of 1300 m a.s.l. with an estimated damaged area of 40% of the total stand.

## 2.2. Machine Details

According to Brunberg [33], different forwarders can be grouped by loading capacity from light forwarders (up to 10 t), medium forwarders (from 10 t to 14 t) to heavy forwarders (over 14 t). The forwarders used in the three different sites included two medium-size forwarders, a John Deere 1210 E (Site A) and an Ecolog 574 B (Site B), with, respectively, 136 and 129 kW engine power, and one heavy forwarder (Site C), Ponsse Buffalo, with 210 kW engine power. Both the John Deere 1210 E and Ponsse Buffalo were equipped with a synchronized winch, but during field activities the winch was only used in Site C. Details of the machines are reported in Table 2.

**Table 2.** Machine details for the different harvesting areas.

| Case Studies | | Site A | Site B | Site C |
|---|---|---|---|---|
| Model | | John Deere 1210 E | Ecolog 574 B | Ponsse Buffalo |
| Engine | - | John Deere 6068 | CAT 3065 E | Mercedes-Benz MTU OM936 |
| Power | kW (hp) | 136 (183) | 129 (173) | 210 (286) |
| Ground clearance | mm | 605 | 650 | 680 |
| Cylinders | n° | 6 | 6 | 6 |
| Wheel number | n° | 8 | 8 | 8 |
| Steering angle | ° | 44 | 40 | 44 |
| Weight empty | t | 18.1 | 17.0 | 19.8 |
| Load capacity | t | 13 | 14 | 15 |
| Tire size | - | 710/45-26,5 | 710/45-26,5 | 710/45-26,5 |
| Boom crane model | - | CF7 | Loglift 83 | Ponsse K 90+ M |
| Gross lifting torque | kNm | 125 | 76 | 140 |
| Maximum boom reach | m | 10.0 | 8.4 | 10.0 |

## 2.3. Data Collection

The data collection was based on a time and motion study through video recording using on-board digital video cameras (Drift®, Ghost-HD). The acquisition rate was set at 720 ppm with 30 frame/s and a field of view of 170°, as also proposed in similar studies [24,34–36]. As proposed by Mologni et al. [34], in order to estimate forwarder load volume, high-resolution photos were taken of each load both from the back and laterally. Photos were then corrected in terms of perspective correction using GIMP® software (GNU Image Manipulation Program) to reduce minimal distortions and scaled by AutoCAD® according to the measured forwarder bunk width.

The machine position was collected using a GNSS receiver (Garmin® 64S) located inside the cabin. For better reception, the GNSS was integrated with an external magnetic antenna (Garmin® 25MCX) located outside the cabin. Both the video cameras and the GNSS receiver were powered with a 20,000 mAh power bank. Machine inclination was detected using a high-frequency three-axis accelerometer installed on the chassis of the forwarder (data acquisition set at 25 Hz) and powered by its autonomous battery. Terrain slope was estimated using QGIS® software in order to compute the slope calculation from the digital elevation model (DTM), in all the sites, a DTM with a resolution of 1 m was available.

A dedicated R code was developed to analyze the data. The extraction distance was estimated for each work cycle using the GNSS data and applying a position filter of 5 m. The machine inclination was determined by the accelerometer data through a specific code developed in R software, with a resolution of 1 Hz.

### 2.4. Time and Motion Study and Work Phase Classification

A time-motion study was carried out through a stop-watch method which was based on video analysis and was set to time units of 1 s (sexagesimal). As common in time-study, observed time was separated into work time (WT) and non-work time (NT) [37]. WT included the productive work time, the related main working time, and the complementary working time, while NT was separated into mechanical, operational, and personal delays as proposed by different authors [22,38].

Forwarder operations from the felling site to the landing site are characterized by a cyclic work (namely work cycle or also turn) which can be divided into the following work elements:

1.  travel empty: movement from the landing site (empty) to the loading site. Starts from the movement of the wheels, ends with the swing of the boom crane (priority 2);
2.  loading: starts with the swing of the boom crane at the loading site, ends when the boom crane stops to swing (priority 1);
3.  driving while loading: movement from different loading sites, starts from the movement of the wheels at loading site after a loading working element; ends when the boom crane starts to swing (priority 2);
4.  travel loaded: movement from the loading site to landing site; starts with the movement of the wheels, ends when the boom crane starts to swing (priority 2);
5.  unloading: when the boom crane unloads the logs at landing site; starts when the wheels stop and the boom crane starts to swing, end when the boom crane stops to swing and wheels start to move at the end of the unloading phase (priority 1);
6.  delay time: included delays up to 15 min.

The time-motion study analyzed the working time according to the previous work element classification. If work elements were overlapping, the element with the highest priority (lower number) was recorded as also proposed by Manner et al. [39].

### 2.5. Independent Variables

A total of 13 explanatory variables for productivity model were investigated (Table 3). For each cycle, the following independent variables were defined: total travel distance and average speed (derived by the data collected on the machine position), maximum and average machine inclination (derived by the data recorded through a three-axis accelerometer); maximum and average machine inclination (derived by GIS analysis), load volume (over bark), and total log number (both derived by the analysis of the photos of each forwarder loaded volume). In the present study, the inclination of the machine was calculated as the resulting inclination between the roll and the pitch of the machine itself. This choice was motivated by the fact that the forwarders work continuously in conditions such that they were inclined both horizontally and sideways. For this reason, the inclination resulting from the roll and pitch represents a unique absolute value of inclination of the machine.

The average ground slope, average machine inclination, maximum ground slope, and the maximum machine inclination were assigned in relation to the following work elements: moving (including travel empty and travel loaded) and loading time. The average speed per cycle (speed) was estimated dividing the travel distance for each cycle for the respective moving time (sum of travel empty, drive while loading, and travel loaded for each cycle).

**Table 3.** Description of the variables tested in the time models.

| Definitions | Variable |
|---|---|
| Average ground slope during moving time | $AGS_M$ |
| Average ground slope during loading time | $AGS_L$ |
| Average log volume | $ALV$ |
| Average machine inclination during moving time | $AMI_M$ |
| Average machine inclination during loading time | $AMI_L$ |
| Load volume | $LV$ |
| Maximum ground slope during moving time | $MGS_M$ |
| Maximum ground slope during loading time | $MGS_L$ |
| Maximum machine inclination during moving time | $MMI_M$ |
| Maximum machine inclination during loading time | $MMI_L$ |
| Number of logs | $NL$ |
| Speed | $S$ |
| Travel distance | $TD$ |

### 2.6. Productivity Model and Statistical Analysis

The productivity model, considering delay-free productive machine hour ($PMH_0$) was defined using all the data from Sites A, B, and C with the work cycle set as the observational unit. Time-consumption models were divided into the following different equations:

1.  Time moving (TM): the sum of travel empty, travel while loading, and travel loaded in minutes (Equation (1));
2.  Time loading (TL): loading time in minutes (Equation (2));
3.  Time unloading (TU): unloading time in minutes (Equation (3)).

To convert the delay-free productive machine hour ($PMH_0$) into productivity machine hour including delays shorter than 15 min ($PMH_{15}$), a correction factor $k$ should be applied. The $k$ factor can be derived from the study observations if the study is based on a long observation time. In the present study, the observation time is not long enough, and it is suggested to derive the $k$ factors from the literature. In the present study, as proposed by Holzleitner et al. [35], a correction factor of 0.3 was used as indicated by Stampfer [40]. In order to estimate total time consumption per cycle (TTC), including delays less than 15 min and expressed in hours, the sum of TM, TL, and TU (Equations (1)–(3)), multiplied by $k$ factor, were divided by 60 (Equation (4)).

As proposed by Tiernan et al. [20], predicted productivity can be estimated by the total time consumption (TTC) for extracting know load volume (LV) as expressed in Equation (5).

$$TM = f(AGS_M,\ ALV, AMI_M,\ LV,\ MGS_M, MMI_M,\ NL,\ TD, S) \tag{1}$$

$$TL = f(AGS_L,\ ALV, AMI_L,\ LV,\ MGS_L, MMI_L,\ NL) \tag{2}$$

$$TU = f(ALV,\ LV,\ NL) \tag{3}$$

$$TTC\ (PMH_{15}) = \frac{60}{(1+k) * (TM + TL + TU)} \tag{4}$$

$$Productivity\left(\frac{m^3}{PMH_{15}}\right) = \frac{LV}{TTC} \tag{5}$$

Due to the diversity of the site areas (e.g., working conditions, operator's experience, type and technology of the forwarders) the site variable was assumed as random factor and the regression analysis was based on random-intercept linear mixed effect models [41,42].

As indicate by Hiesl et al. [42], the random intercept included the different influences of the operator, machine, and site conditions at each location. Random intercept mixed effect models consider this different combination of operator, machine, and site at each location as unique. Random slope of

the regression was also tested for the significance as appropriate. A likelihood ratio test was used to evaluate the significance of the individual variables; the significance level of the statistical analysis was set to 0.05. In order to analyze statistical assumptions, residuals were checked using residual plot distributions. In the case of non-normal distribution of the residual, logarithmic and square root transformations were tested on both response variables and explanatory continuous variables. The goodness of fit of linear mixed effect models was tested through the coefficient of determination ($R^2_{LR}$) proposed by Magee [43] and based on likelihood ratio joint significance.

## 3. Results

Total time studies covered 57.90 $PMH_{15}$, divided into 10.41 $PMH_{15}$ in Site A, 23.64 $PMH_{15}$ in Site B, and 23.85 $PMH_{15}$ in Site C. Field activities covered 10 days, three days in both Site A and Site B, and four days in Site C. Within this time, the forwarders completed 101 working cycles, 16 in Site A, 33 in Site B, and 52 in Site C. Total extracted timber volume for all sites was 1277 $m^3$, resulting in 4284 logs. About 223 $m^3$ and 884 logs were extracted in Site A, 417 $m^3$ and 1385 logs in Site B, 637 $m^3$ and 2015 logs in Site C. Time consumption was distributed as shown in Figure 2. The mean time consumption per cycle was 38.7, 42.2, and 25.1 $PMH_{15}$, respectively for Sites A, B, and C.

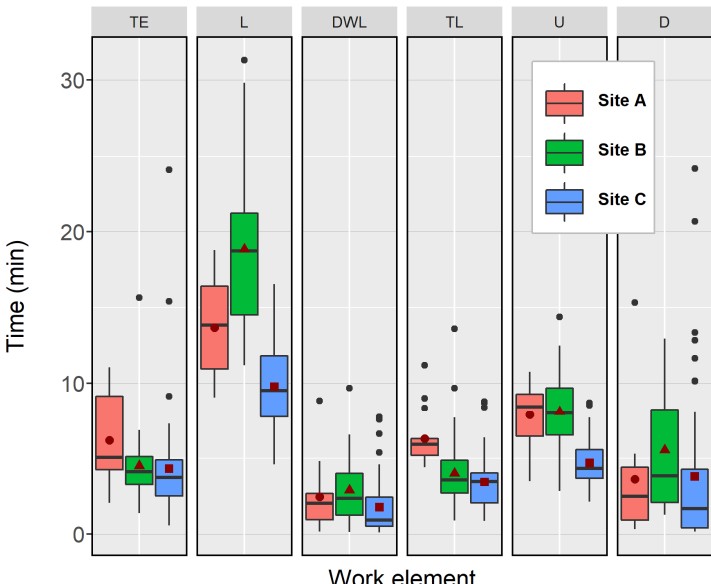

**Figure 2.** Time consumption of work element for different site where TE is travel-empty, L is loading, DWL is drive while loading, TL is travel loaded, U is unloading, and D is delay. The boxes include the variability of the data between the 25th and the 75th percentiles. The horizontal black line represents the median while the circle, triangle pointing upwards, and square in dark red represent the mean.

Average productivity was 22.5 $m^3$/$PMH_{15}$ in Site A, 18.5 $m^3$/$PMH_{15}$ in Site B, and 29.4 $m^3$/$PMH_{15}$ in Site C. As shown in Figure 3, especially in Site C there was higher variability, due to the different types of assortments extracted and the use of the winch on the forwarder.

Higher mean time consumption per cycle was recorded in Site B, probably due to the highest mean ground slope and machine inclination (Table 4). The longest work element was loading phase for all sites. In particular, loading phase in Site B was higher than Sites A and C, probably due to the highest average number of logs, average load volume, and corresponding average log volume (Table 4). Furthermore, the forwarder used in Site B was a medium-size machine with lower power compared with the others and not equipped with self-levelling cabin, and the working area was completely covered by windthrows. Loading time was lower in Site C, where the average number of logs extracted was lower and the corresponding average log volume was higher.

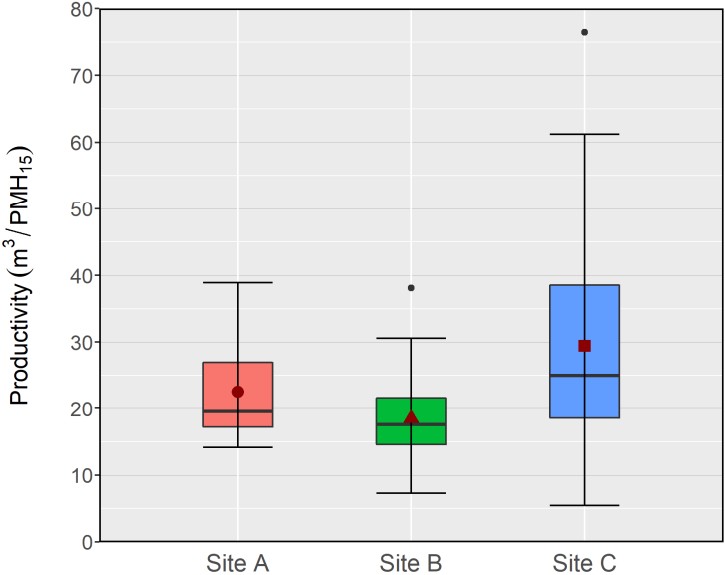

**Figure 3.** Variability of forwarder productivity for the three sites with work cycle as observational unit. The boxes include the variability of the data between the 25th and the 75th percentiles. The horizontal black line represents the median while the circle, triangle pointing upwards, and square in dark red represent the mean.

**Table 4.** Descriptive statistics for the independent variables considered in the statistical analysis.

| Variables | Unit | Site A | | | Site B | | | Site C | | |
|---|---|---|---|---|---|---|---|---|---|---|
| | | Min | Average | Max | Min | Average | Max | Min | Average | Max |
| $AGS_M$ | % | 11.2 | 15.7 | 22.8 | 9.5 | 24.7 | 32.7 | 17.6 | 31.2 | 43.9 |
| $AGS_L$ | % | 11.5 | 23.1 | 36.1 | 9.5 | 25.1 | 34.7 | 19.3 | 34.1 | 72.6 |
| ALV | $m^3$ | 0.1 | 0.34 | 0.67 | 0.15 | 0.33 | 0.81 | 0.11 | 0.36 | 0.81 |
| $AMI_M$ | % | 18.0 | 22.6 | 26.5 | 21.6 | 30.7 | 38.4 | 19.9 | 39.6 | 57.6 |
| $AMI_L$ | % | 15.0 | 21.5 | 39.7 | 7.5 | 33.4 | 60.1 | 17.0 | 41.7 | 65.0 |
| LV | | 10.1 | 13.9 | 19.6 | 6.3 | 12.7 | 18.7 | 1.9 | 12.2 | 24.9 |
| $MGS_M$ | % | 26.9 | 47.4 | 85.9 | 20.9 | 43.3 | 68.9 | 26.2 | 45.3 | 85.5 |
| $MGS_L$ | % | 24.3 | 43.9 | 85.9 | 19.2 | 31.9 | 56.7 | 24.8 | 40.4 | 94.8 |
| $MMI_M$ | % | 20.2 | 43.6 | 57.3 | 43.0 | 66.4 | 88.6 | 38.8 | 70.0 | 96.2 |
| $MMI_L$ | % | 20.3 | 38.0 | 58.8 | 20.5 | 56.9 | 90.4 | 31.6 | 65.6 | 95.6 |
| NL | n° | 26 | 55.0 | 104 | 23 | 42 | 65 | 11 | 38 | 86 |
| S | $m\ s^{-1}$ | 1 | 1.3 | 1.6 | 0.4 | 0.8 | 1.3 | 0.3 | 0.8 | 1.7 |
| TD | m | 606 | 1181 | 1834 | 217 | 557 | 1457 | 41 | 434 | 2221 |

Note: $AGS_M$: average ground slope during moving; $AGS_L$: average ground slope during loading; ALV: average log volume; $AMI_M$: average machine inclination during moving; $AMI_L$: average machine inclination during loading; LV: load volume ($m^3$ are over bark); $MGS_M$: maximum ground slope during moving; $MGS_L$: maximum ground slope during loading; $MMI_M$: maximum machine inclination during moving; $MMI_L$: maximum machine inclination during loading; NL: number of logs; S: speed; TD: travel distance.

In order to detect and estimate the variables affecting forwarder productivity in salvage logging operations, the variables in Table 5 were used in the linear mixed effect models. The highest speed (S) was registered in Site A, where the respective travel distance was the highest and the average ground slope and machine inclination during moving and loading were lower.

**Table 5.** Explanatory variables of fixed effect in time moving (TM), time loading (TL), and time unloading (TU) (Log: the logarithmic transformation of the variable).

| Equation | Response Variable | Coefficient | Estim. | SE | *t* Value | *p* Value |
|---|---|---|---|---|---|---|
| 1 (TM) | Time moving (min) | ALV (m$^3$) | −4.151 | 1.28 | −3.240 | 0.010 |
| | | MGS$_M$ (%) | 0.075 | 0.02 | 3.764 | <0.001 |
| 2 (TL) | Log Time loading (min) | ALV (m$^3$) | −1.004 | 0.134 | −7.516 | <0.001 |
| | | MMI$_L$ (%) | −0.006 | 0.001 | −4.344 | <0.001 |
| | | MGS$_L$ (%) | 0.005 | 0.002 | 3.192 | 0.002 |
| 3 (TU) | Log Time unloading (min) | LV (m$^3$) | 0.041 | 0.007 | 5.965 | <0.001 |
| | | NL (n°) | −0.004 | 0.002 | −2.324 | 0.020 |

Note: ALV: average log volume; MGS$_M$: maximum machine ground slope; MMI$_L$: maximum machine inclination during loading; MGS$_L$: maximum ground slope during loading; LV: load volume (over bark); NL: number of logs.

*Productivity Equations*

The aim of this study was to improve the understanding of time consumption and forwarder productivity salvage logging operations and in difficult terrain, and not to compare the productivity among different sites. Response variables of the time moving (TM) (Table 5) were the average log volume (ALV), max ground slope (MGS$_M$), and travel distance (TD). ALV had a negative influence on the time moving, while MGS$_M$ and TD had a positive influence on the time moving. Therefore, with an increase of ALV, time moving decreased. On the contrary, with an increase of MGS$_M$ or TD, time moving increased.

As the slope of explanatory variables, in particular ALV and TD, would seem to vary among the site areas, both random intercept and slope were tested. The highest significance of Equation (1) (TM) was reached using distance as random slope (Table 6).

**Table 6.** Explanatory variable of random effect using in Equation (1) TM.

| Equation | Coefficient | SD | Variance | *p* Value |
|---|---|---|---|---|
| 1 (TM) | TD | 0.002 | −228.5 | <0.001 |

Note: TD: Travel distance.

The estimate of the random slope of the TD was 0.07 for Site A, 0.09 for Site B, and 0.01 for Site C. The random slope showed a significant influence between the equations (*p* value 0.014). Equation (1) (TM) (Table 7) explains over 75% of the variability.

**Table 7.** Random intercepts and goodness of fit of the linear mixed-effect models.

| Equation | N | Variance | SD | L$_M$ | L$_0$ | R$^2$$_{LR}$ |
|---|---|---|---|---|---|---|
| 1 (TM) | 101 | 4.118 | 2.029 | −228.49 | −295.50 | 0.735 |
| 2 (TL) | 101 | 0.07 | 0.268 | 6.389 | −19.56 | 0.402 |
| 3 (TU) | 101 | 0.06 | 0.249 | −13.12 | −32.64 | 0.321 |

Note: N: number of observations; L$_M$: log likelihoods of the model; L$_0$: log likelihoods of the intercept; R$^2$$_{LR}$: coefficient of determination proposed by Magee [43].

The variables that significantly affect time loading (TL–Equation (2)) (Table 5) were the average log volume (ALV), maximum machine inclination (MMI$_L$), and maximum ground slope (MGS$_L$). Equation (2) explains over 40% of the variability, where ALV and MMI$_L$ had a negative influence on the time loading while MGS$_L$ had a positive effect on time loading. Anyway, the effect of MMI$_L$ and of MGS$_L$ on the loading operation is marginal if it is compared to the higher effect of the ALV. The presence of single obstacles along the corridors, such as stumps or large stones, can affect in some

cases the position of the machine during the loading operation as well as the maximum ground slope along the extraction trail.

Equation (3) related to time unloading (TM) (Table 7) explains the 32% of variability, variables that significantly affect time unloading were load volume (LV) and number of logs (NL). LV had a positive influence on time unloading while NL had a negative influence on time unloading (Table 5).

Figure 4 shows the effect of the average log volume and travel distance on productivity, according to the different equations applied. In Sites A and C, forwarder productivity results were slightly similar over 600 m of travel distance. Furthermore, over 800 m travel distance, with an ALV of 0.3 and 0.35 $m^3$ in Site A and an ALV of 0.25 and 0.30 $m^3$ in Site C, Site A and Site C productivity overlapped.

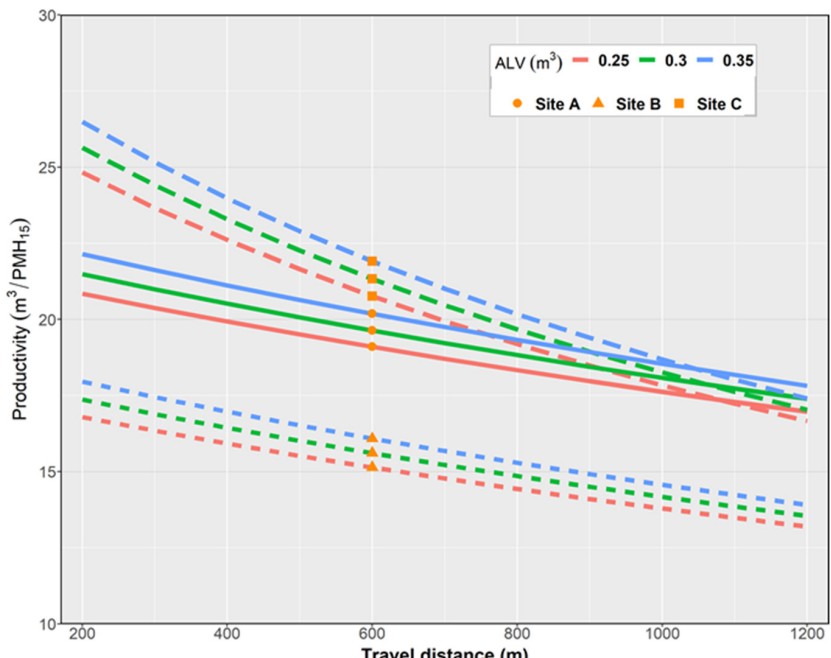

**Figure 4.** Predicted forwarder productivity ($m^3$/PMH$_{15}$) in relation to the total travel distance with ALV (average log volume) of 0.25, 0.3, and 0.35 $m^3$. The models, based on linear mixed effect models, consider site variables as random factor. Triangle pointing upwards, circles, and squares represent, respectively, Sites A, B, and C.

## 4. Discussions

In the current study related to salvage logging operations in difficult terrain, the predicted productivity of forwarder shows a range from 14.4 to 20.5 $m^3$/PMH$_{15}$ for a medium-size machine (10–14 t) and from 18.8 to 23.0 $m^3$/PMH$_{15}$ for a heavy-size machine (over 14 t) with an extraction distance between 250 and 500 m.

Forwarder productivity is generally affected mainly by load volume and the extraction distance, as it is clearly reported in [20,27,28]. Additionally, in the current study in salvage logging condition, the extraction distance has a significant influence on forwarder productivity.

The predicted productivity, with a total travel distance of 500 m resulted in 20.5, 16.3, 23.0 $m^3$/PMH$_{15}$, respectively, for the forwarder operating in Sites A, B, and C. Overall, an increase of 200 m of total travel distance reduces the productivity by 6% on average.

For what it concerns the lower productivity of the forwarder reported in Site B, this could be affected by the type of machine (medium-size forwarder without self-levelling and rotating cabin) as well as by the lower experience of the operator as was also identified by [25,44,45]. The forwarder operator working at Site B, in fact, had six months experience, while the operator on forwarder operating in Site A had 10 years and the operator on the forwarder operating in Site C had about two years of experience.

In this study, the combination between machine characteristics, machine operator's experience, as well site conditions were unique at each location. As a consequence, the adaptation of the linear mixed-effect model was used to define a random intercept representing the combination of the previous variables as also suggested for a similar case by [42].

Figure 5 compares (up to a maximum travel distance of 1200 m) the productivity model of the present study with the productivity models gathered from other references. In this case, to keep consistent the $m^3/PMH_{15}$ between the different productivity models a delay factor (*k*) of 0.3 [40] was applied.

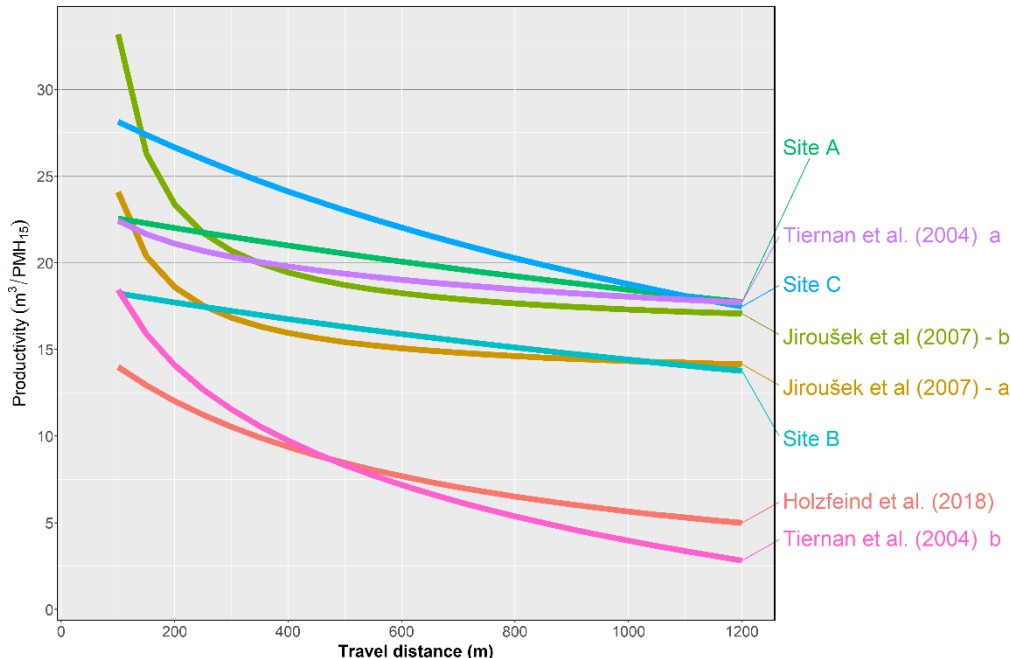

**Figure 5.** Comparison of variation of forwarder productivity with total travel distance with previous publications.

In Sites A and C, the productivity of the forwarders was slightly similar to Tiernan et al. (2004) (model a) [20], where the model explains the effect of travel distance on the productivity in easy terrain conditions (with even ground conditions and terrain slopes less than 10%) in clearcutting operations. Tiernan et al. (2004) (model b) [20] shows the effect of the travel distance on forwarder productivity in difficult terrain conditions (with poor ground conditions and ground slope greater than 10%) during clearcutting operations. The average extraction distance ranges from 100 to 700 m estimated by Tiernan et al. [20]. In order to compare extraction distances estimated by Tiernan et al. [20] (estimated as half of the travel distance per cycle) with the travel distance in this study (estimated as travel distance per cycle) the extraction distance estimated by Tiernan et al. [20] is doubled. In addition, Tiernan et al. [20] models to predict forwarder productivity in clearcut areas includes a total of 56 cycles. Jiroušek et al. [29] reports two models (a and b) explaining, respectively, the effect of travel distance on the productivity with medium and heavy forwarders, where the medium forwarder was a forwarder with a payload from 10 to 12 t and the heavy forwarder was a forwarder with a payload higher than 12 t. The productivity of medium forwarder is similar to medium forwarder of Site B, while the productivity of the heavy forwarder compares well with the Site A medium forwarder over 200 m of travel distance.

Holzfeind et al. [24] estimate productivity of 16 $m^3/PMH_{15}$ for a John Deere 1110E medium-size winch assist forwarder with an average log volume of 0.22 $m^3$, an average extraction distance of 111 m, an average extraction slope of 29.2%, and a load volume of 9.25 $m^3$, where the average extraction distance was estimated as the distance from the closest loading point to the unloading point. In the current study, the heavy winch assist forwarder operating in Site C reported a higher productivity

than [22]. This is probably due to larger size of the machine as well the higher concentration of logs along the corridors as it is common on salvage logging operations. This circumstance is consistent with [30], reporting that the forwarder productivity is similar to the productivity in normal clearcuts.

## 5. Conclusions

The present study analyzed and evaluated the productivity of forwarders in salvage logging operations and identified the most significant variables affecting productivity in difficult terrain. This study reports that the productivity in salvage logging in difficult terrain with a heavy forwarder (Site C) is in line with forwarder productivity in easy conditions during clearcut operations. Predicted productivities are positively affected by average load volume, maximum machine inclination during loading, and number of logs. On the contrary, travel distance and maximum ground slope have a strong negative influence on productivity. The results of the models confirm the effect of the travel distance on forwarder productivity and could be used in the future to plan and optimize road distribution in order to optimize forwarding efficiency in salvage logging operations based on CTL systems. In particular, the results partially cover the lack of knowledge of forwarder productivity in salvage logging operations and represent the first study of forwarder productivity in large scale windthrow and in difficult terrain in the Alps. In addition, the results quantified the variation of forwarder productivity with average log volume, ground slope, travel distance, machine inclination, load volume, and number of logs. In terms of road distribution, the position of these roads should be such as to minimize the extent of forwarding that may be economical. Although in this study machine inclination significantly affected forwarding productivity, higher machine inclination could also affect mechanical part of the machines.

However, forwarder productivity could be also related to harvester productivity and efficiency during salvage logging operation in difficult terrain. In order to better understand efficiency and the productivity of a CTL systems in non-ordinary conditions, such as salvage logging operations, further investigations involving harvester and forwarder operations are necessary.

**Author Contributions:** Conceptualization, A.C., O.M. and S.G.; methodology, A.C., O.M. and S.G.; software, A.C. and O.M.; validation, A.C. and O.M.; formal analysis, A.C., O.M. and S.G.; investigation, A.C. and O.M.; resources, A.C. and O.M.; data curation, A.C.; writing—original draft preparation, A.C. and S.G.; writing—review and editing, A.C., O.M., D.R., R.C. and S.G.; visualization, A.C.; supervision, S.G.; project administration, S.G; funding acquisition, R.C. and S.G. All authors have read and agreed to the published version of the manuscript.

**Funding:** This study is part of the CARE4C project found by EU Commission (GA 778322) and of the Young research for VAIA of the PhD LERH Program of the Università degli Studi di Padova in the frame of VAIA-Front project of TESAF Department.

**Acknowledgments:** We thank the contractors involved in this study, the office of forest planning of Bolzano province, the forest technicians of Levico Terme TN and Belluno, and the master's degree candidates B.Sc. Giovanni Bellan and B.Sc. Gaetano D'Anna.

**Conflicts of Interest:** The authors declare no conflict of interest.

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
