# Peer review of "Forwarder Productivity in Salvage Logging Operations in Difficult Terrain"

_forests, doi:10.3390/f11030341_

Round 1
Reviewer 1 Report
This paper aims to partially fill the lack of knowledge of forwarder productivity in 81 salvage logging operation in difficult terrain.
The specific objectives of the paper are:
i) to estimate 82 forwarder productivity in salvage logging in difficult terrain and
ii) to identify significant variables 83 affecting productivity.
However, the aim of the study is too general to publish the manuscript. There are a lot of machine productivity papers to improve machine productivity and operational efficiency. However, this paper only considered forwarder productivity and It is difficult to find any solution to improve the harvesting system or machine selection on different terrain conditions.
Additionally, this paper investigated productivity along with the different terrain conditions. they collected data from three different sites and three different machines and capacity. In forest operation, machine capacity and machine operator skills have highly impacted on machine productivity. However, they did not consider these factors when they developed the productivity model. If you want to test the productivity difference among the different study sites, The experimental environment shall be kept the same condition as possible as you can.
For these reasons, the manuscript needs to rewrite with improved experimental design and objectives.
Author Response
On behalf of all the authors, I ‘d like to thank the reviewer for the appropriate and useful suggestions and comments that have been proposed on the manuscript “Forwarder productivity in salvage logging operations in difficult terrain”. The manuscript has been now revised and the following point-by-point replay is detailed:
Reviewer 1 Comment [1]
This paper aims to partially fill the lack of knowledge of forwarder productivity in salvage logging operation in difficult terrain. The specific objectives of the paper are:
- i) to estimate forwarder productivity in salvage logging in difficult terrain and
- ii) to identify significant variables affecting productivity.
However, the aim of the study is too general to publish the manuscript. There are a lot of machine productivity papers to improve machine productivity and operational efficiency. However, this paper only considered forwarder productivity and It is difficult to find any solution to improve the harvesting system or machine selection on different terrain conditions.
Authors’ replay to Comment [1]
We thank the Reviewer for this comment. We agree on the presence of several papers dealing with forwarder productivity. In fact, some papers have recently been published which present results relating to ordinary forest operation conditions and almost exclusively in uneven terrain. Only a few papers eg. [1,2] are related to harvesting productivity and cost in salvage logging operation but also in these cases are related to uneven terrain.
The recent bibliography does not report studies on the use of these machines in salvage logging operations in difficult terrain. Our papers are specifically related to productivity in salvage logging operations in difficult terrain and based on the forwarder productivity under real working conditions. For these reasons, the aims of the paper are specific for this type of condition which has become quite common in recent years especially in Central and Eastern Europe. In addition, we clarify the specific objectives in the revised paper upload form row 92 to row 95.
Reviewer 1 Comment [2]
Additionally, this paper investigated productivity along with the different terrain conditions. they collected data from three different sites and three different machines and capacity. In forest operation, machine capacity and machine operator skills have highly impacted on machine productivity. However, they did not consider these factors when they developed the productivity model. If you want to test the productivity difference among the different study sites, the experimental environment shall be kept the same condition as possible as you can.
Authors’ replay to Comment [2]
We agree with the comment of the reviewer that the three salvage logging operations consider in our paper are performed by different forwarders which are operated by different operators with different skills and in sites in which ground conditions are different.
For these reasons, the time consumption was analyzed on linear mixed models in which variables of the sites were assumed as a random factor; the regression analysis was thus based on random-intercept in the linear mixed effect model [3,4]. According to Hiesl et al.[3], “The random intercept α consisted of the influences of the operator, machine, and site conditions at each location. This combination of machine, operator, and site conditions was unique at each location”.
The approach of the present paper is different from e.g. Proto et al. [5] in which a linear model was adopted with different site areas which were identified as block factors. In Proto et al. [5] the machines and sites involved were different in terms of size, operator skills and ground conditions, the aims of that study were to analyze individually forwarder productivity and factors affecting forwarder. Also Ghaffarian et al. [6] developed a productive model for forwarder based on 82 working cycles from two different forwarders in terms of size, with different operators skill and in two different sites.
Our paper proposes a different statistic approach and analyzes forwarder productivity under salvage logging operations under real working conditions. According to the assumption of the linear mixed models described above, the operators' experience, the site conditions and the machine models were included in the random intercept by assuming site variable as a random factor in the random intercept mixed effect models. We included some revisions in our manuscript in rows from 195 to 203.
References
- Kärhä, K.; Anttonen, T.; Poikela, A.; Palander, T.; Laur, A. Evaluation of Salvage Logging Productivity and Costs in Windthrown Norway Spruce-Dominated Forests. Forests 2018, 9, 280.
- Bergkvist, I. Beskrivning och Analys av de Dominerande Maskinsystemen (Salvaging Windfalls) – Description and Analysis of Dominant Machine Systems. Skogforsk: Uppsala, Sweden 2005, 598, 15p.
- Hiesl, P.; Benjamin, J.G. A multi-stem feller-buncher cycle-time model for partial harvest of small-diameter wood stands. Int. J. For. Eng. 2013, 24, 101–108.
- Bates, D.; Mächler, M.; Bolker, B.M.; Walker, S.C. Fitting linear mixed-effects models using lme4. J. Stat. Softw. 2015, 67.
- Proto, A.R.; Macrì, G.; Visser, R.; Harrill, H.; Russo, D.; Zimbalatti, G. Factors affecting forwarder productivity. Eur. J. For. Res. 2018, 137, 143–151.
- Ghaffarian, M.R.; Stampfer, K.; Sessions, J. Forwarding productivity in Southern Austria. Croat. J. For. Eng. 2007, 28, 169–175.
Reviewer 2 Report
The issue addressed in the paper discusses the forwarder's productivity in salvage logging operations in difficult terrain. This is an important and interesting research topic. It has been rightly noted that fully mechanized harvesting systems are the preferred solution on trafficable terrains and proper slopes. In the proposed scope, the paper is generally is generally well prepared. However, I recommend a few corrections to improve the quality of this article:
- to discuss the current state of the art (I also suggest a more dilligent description of the research methods; concise, coherent research scenario, current references to be completed etc.);
- to increase the readability of data in tables (e.g. numerical values should be arranged correctly in figures; precise, clear descriptions needed);
- in addition, I advise the Authors to justify the construction of the productivity model (precisely, simply and to explain why no other alternative approaches have been used);
that is, supplement the summary descriptive analysis (please complete point 4).
I also strongly suggest that recommendations for specific, practical, not only general (and not entirely clear) applications of this research shall be provided (please complete point 5).
The language of this paper is relatively correct, however some descriptions would benefit from being more concise (please include native speaker verification).
Author Response
On behalf of all the authors, I ‘d like to thank the reviewer for the appropriate and useful suggestions and comments that have been proposed on the manuscript “Forwarder productivity in salvage logging operations in difficult terrain”. The manuscript has been now revised and the following point-by-point replay are detailed:
Reviewer 2 Comment [1]
Discuss the current state of the art (I also suggest a more diligent description of the research methods; concise, coherent research scenario, current references to be completed, etc.).
Authors’ replay to Comment [1]
We agree with this comment, we improve the state of arts about forwarder productivity from row 73 to row 81. Our paper is related to specific harvesting conditions and there is a lack of knowledge of forwarder productivity in salvage logging operations, in particular in difficult terrain. We also clarify research methods and statistical analysis from row 195 to row 203
Reviewer 2 Comment [2]
Increase the readability of data in tables (e.g. numerical values should be arranged correctly in figures; precise, clear descriptions needed).
Authors’ replay to Comment [2]
According to the comment of the reviewer, we improve the caption of the tables and figures. We also improve the description of the figure and table in order to clarify and make better the readability of the tables and figures.
Reviewer 2 Comment [3]
I advise the Authors to justify the construction of the productivity model (precisely, simply and to explain why no other alternative approaches have been used).
Authors’ replay to Comment [3]
The statistics Time consumption of three different forwarders were analyzed with linear mixed models in which variables of the sites were assumed as a random factor; the regression analysis was thus based on random-intercept in the linear mixed effect model [1;2]. According to Hiesl et al.[1], “The random intercept α consisted of the influences of the operator, machine, and site conditions at each location. This combination of machine, operator, and site conditions was unique at each location”, we take in consideration the variability of ground conditions, operators experience and type of machines to test the significance of the independent variable on time consumption and to develop forwarder productivity model based on random intercept linear mixed effect models. In contrast to linear models, that are also widespread in productivity studies, this type of model quantifies the variability between working areas by the coefficient of the random intercept. We add some specific justification about the construction and type of model used in our paper from row 195 to raw 203.
Reviewer 2 Comment [4]
I also strongly suggest that recommendations for specific, practical, not only general (and not entirely clear) applications of this research shall be provided (please complete point 5)
Authors’ replay to Comment [4]
We add more specific and practical applications of this study from row 327 to row 334. Salvage logging operations is increasing in Central Europe because of the growth of severe atmospheric events, this study is the first one about this specific topic in European Alps (salvage logging productivity in difficult terrain) and this paper want to partially cover the lack of knowledge quantifying the variation of forwarder productivity and the variables that affecting forwarder productivity.
References
- Hiesl, P.; Benjamin, J.G. A multi-stem feller-buncher cycle-time model for partial harvest of small-diameter wood stands. Int. J. For. Eng. 2013, 24, 101–108.
- Bates, D.; Mächler, M.; Bolker, B.M.; Walker, S.C. Fitting linear mixed-effects models using lme4. J. Stat. Softw. 2015, 67.

Round 2
Reviewer 1 Report
The authors try to answer the previous comments and I agree and accept their contribution to these research outputs.
Happy to accept this paper on Forests Journal.
Best regards,